# DIRAS3 Inhibits Ovarian Cancer Cell Growth by Blocking the Fibronectin-Mediated Integrin β1/FAK/AKT Signaling Pathway

**DOI:** 10.3390/cells14161250

**Published:** 2025-08-13

**Authors:** Jing Guo, Janice M. Santiago-O’Farrill, Vivian Orellana, Rumeysa Ozyurt, Hailing Yang, Marc Pina, Gamze Bildik, Weiqun Mao, Robert C. Bast, Zhen Lu

**Affiliations:** 1Department of Obstetrics and Gynecology, Ruijin Hospital Affiliated to Shanghai Jiao Tong University School of Medicine, Shanghai 200025, China; jguo12@foxmail.com; 2Department of Experimental Therapeutics, University of Texas MD Anderson Cancer Center, Houston, TX 77054, USA; jmsantiago@mdanderson.org (J.M.S.-O.); vrorellana@mdanderson.org (V.O.); rozyurt@mdanderson.org (R.O.); hyang3@mdanderson.org (H.Y.); mapina@manderson.org (M.P.); gbildik@mdanderson.org (G.B.); wmao@mdanderson.org (W.M.)

**Keywords:** ovarian cancer, autophagy, fibronectin, integrin, FAK

## Abstract

Autophagy is a crucial cellular process responsible for sustaining homeostasis through the degradation and recycling of proteins and organelles, providing energy during amino acid starvation and hypoxia. In cancer, autophagy can either inhibit tumor growth or support cancer cell survival. Our previous studies have shown that re-expression of the tumor suppressor gene *DIRAS3* inhibits growth of ovarian cancer cells, promotes autophagic cell death in vitro, and induces tumor dormancy in vivo. Growth factors and extracellular matrix (ECM) components can, however, inhibit DIRAS3-induced autophagic cell death. This study explores whether fibronectin (FN) can counteract the growth inhibition induced by DIRAS3 in ovarian cancer cells. FN was found to inhibit DIRAS3-induced autophagy and to partially rescue ovarian cancer cells from DIRAS3-induced cell death while reducing DIRAS3-induced inhibition of p-FAK and p-AKT. Inhibiting FAK with defactinib in ovarian cancer cells enhanced DIRAS3-induced autophagy and cell death. Re-expression of DIRAS3 and treatment with defactinib produced tumor regression in xenograft models. Our findings suggest that ECM components in the tumor microenvironment like FN enhance the activities of β1 integrin, FAK, and AKT to inhibit DIRAS3-induced autophagic cell death, thereby promoting ovarian cancer cell survival.

## 1. Introduction

Despite the significant progress in surgery and chemotherapy, ovarian cancer continues to be a leading cause of mortality in gynecologic cancers, with an estimated 20,890 new cases and 12,730 deaths expected in the United States in 2025 [1]. A major factor contributing to poor outcomes in ovarian cancer is the persistence of dormant, drug-resistant cells after standard treatment with surgery, carboplatin and paclitaxel. Even when CA125 levels normalize and imaging results appear negative, exploratory “second look” surgery can document small, poorly vascularized nodules of residual disease on the peritoneal surface in approximately 50% of patients [2,3].

Persistent ovarian cancer cells from positive second look operations upregulate DIRAS3 and undergo autophagy [4]. DIRAS3 is an imprinted tumor suppressor gene with significant homology to RAS that is expressed in normal ovarian and fallopian tube cells but downregulated in approximately 60% of ovarian cancers [5]. DIRAS3 re-expression in ovarian cancer cells suppresses proliferation, motility, and growth as xenografts [6]. Expression of DIRAS3 induces autophagy by multiple mechanisms [7]. Autophagy has been shown to be required for survival of dormant human ovarian cancer cells in a xenograft model [8]. When DIRAS3 was re-expressed in human ovarian cancer xenografts, tumors remained dormant, but when levels of DIRAS3 were reduced, tumors grew rapidly [6]. Treatment of dormant DIRAS3-expressing xenografts with chloroquine, an autophagy inhibitor, reduced tumor growth when DIRAS3 was subsequently reduced [7,9].

Autophagy is a cellular mechanism that helps maintain homeostasis by breaking down and recycling long-lived or misfolded proteins and damaged organelles [10]. This process can protect cells from stress by providing energy to sustain organelle function and cellular survival pathways [10]. Sustained autophagy can, however, be lethal [10]. In the context of cancer, autophagy can play this dual role, either enhancing cancer cell death or sustaining cancer cell survival in a hypoxic, nutrient-poor environment [11,12].

Tumor dormancy is characterized by the persistence of microscopic deposits of metastatic cancer cells that maintain an active balance between cell proliferation and cell death. These cells are not actively cycling and lack effective angiogenesis [13,14]. When growth factors and cytokines are present, dormant cells can be stimulated to actively divide and proliferate, leading to progressive tumor growth. This activation happens through the binding of these signaling molecules to cell surface receptors, triggering and initiating intracellular signaling cascades. These signals can activate oncogenes or inhibit tumor suppressor genes, thereby driving cancer progression [13,14].

Overexpression or remodeling of the extracellular matrix (ECM) is frequently observed in ovarian cancers, leading to the activation of cell proliferation and migration pathways [15]. Fibronectin (FN), an abundant ECM glycoprotein found in many cancers, plays a key role in cancer development by enabling cell survival within FN-containing matrices [16,17,18]. Previously, we documented that IGF-1, VEGF, IL-8, and matrix proteins (collagen, fibronectin, and poly-l-lysine) are found in the xenograft microenvironment and can inhibit DIRAS3-induced autophagic cell death in culture. Interestingly, previous studies indicate that fibronectin sustains survival of ovarian cancer cells [19]. The pro-survival effect of FN is through interaction with integrin β1, which subsequently activates Focal adhesion kinase (FAK) and the AKT signaling pathway [20,21].

Based on the clinical significance of dormant, drug-resistant ovarian cancer cells and the reported role of FN in promoting cancer cell survival, we hypothesized that FN could inhibits DIRAS3-induced autophagic cell death by inhibiting the FAK/AKT pathway, thereby supporting ovarian cancer cells survival. In the current study, we used a model with inducible expression of DIRAS3 in SKOv3 and OVCAR8 ovarian cancer cells to investigate whether FN plays an important role in counteracting the growth inhibition induced by DIRAS3 in ovarian cancer cells. We found that FN prevents DIRAS3-induced autophagic cancer cell death by weakening DIRAS3-mediated inhibition of p-FAK and p-AKT. Inhibiting the FAK signaling pathway enhanced DIRAS3-induced autophagy, leading to ovarian cancer cell death, both in vitro and in vivo.

These data provide insight into the mechanism by which factors in the tumor microenvironment can promote ovarian cancer cell survival by inhibiting DIRAS3-induced autophagic cell death. Importantly, high expression levels of FN1 and ITGB1 are linked with poorer survival in ovarian cancer patients, highlighting the potential clinical importance of this signaling axis [22,23].

## 2. Materials and Methods

### 2.1. Antibodies and Reagents

Antibodies targeting LC3B (#3868 and #2775), p-FAK (#3283), p-AKT (#4060), AKT (#9272), and Integrin β1 (#4706) were obtained from Cell Signaling Technology (Danvers, MA, USA). The GAPDH (sc-32233) antibody was purchased from Santa Cruz Biotechnology (Dallas, TX, USA). Doxycycline hyclate (DOX, D9891) was purchased from Sigma-Aldrich (St. Louis, MO, USA), while the FN (5080) was purchased from Advanced Biomatrix (Carlsbad, CA, USA). The FAK inhibitor, Defactinib (VS-6063, PF-04554878) was purchased from Selleckchem (Houston, TX, USA).

### 2.2. Cell Culture

DIRAS3-inducible SKOv3 cells were cultured in McCoy’s medium containing 10% FBS, 200 µg/mL G418 (Sigma, St. Louis, MO, USA), and 0.12 µg/mL puromycin (Sigma, St. Louis, MO, USA). OVCAR8-DIRAS3 ovarian cancer cells were cultured in Roswell Park Memorial Institute 1640 medium (RPMI-1640) supplemented with 10% FBS, 500 µg/mL G418 (Sigma, St. Louis, MO, USA). Both cell lines were tested for Mycoplasma contamination prior to use. Cells were maintained at 37 °C in a humidified incubator with 5% CO_2_.

### 2.3. Reverse-Phase Protein Arrays (RPPAs)

SKOv3-DIRAS3 cells were treated with DOX (1 ug/mL) to induce DIRAS3 expression and cultured with FN for durations of 16, 24, and 48 h. Cells were then lysed, and the protein extracts were subjected to RPPA analysis to the MD Anderson Cancer Center RPPA Core Facility. The samples were then probed with a validated panel of 161 antibodies against signaling and growth regulatory proteins.

### 2.4. Western Blotting Immunoblot

A total of 150,000 SKOv3-DIRAS3 and OVCAR8-DIRAS3 cells were plated in 6-well plates coated with or without different concentrations of FN. Cells were then treated with DOX for 48 h. Cell lysates were prepared by adding 100 μL of SDS lysis buffer to each well. Cells were scraped, collected, and heated at 99 °C for 5 min. Equal amounts of protein were separated using 8–15% SDS-PAGE, transferred to PVDF membranes, and analyzed by Western blot with an ECL chemiluminescence reagent (GE Healthcare, Chicago, IL, USA). The intensity of the bands was quantified by using ImageJ (free, Java-based).

### 2.5. Immunofluorescent Staining

SKOv3-DIRAS3 (1.8 × 10^5^/well) cells were cultured overnight in 6-well plates coated with or without FN (10 mg/mL) for one hour. Following this, cells were treated with DOX to induce DIRAS3 expression for up to 48 h. After treatment, cover slips were washed four times for 5 min in PBS, mounted on glass slides using Vectashield fluorescent mounting media (Vector Lab, Burlingame, CA, USA) and examined with fluorescence microscopy (Olympus 1X71, Center Valley, PA, USA). Goat anti-rabbit secondary antibodies conjugated to Alexa Fluor 488 were purchased from Thermo Scientific (Waltham, MA, USA).

### 2.6. Transmission Electron Microscopy (TEM)

For TEM examination of autophagosomes, SKOv3-DIRAS3 and OVCAR8-DIRAS3 cells were plated in 6-well plates and treated with DOX, followed by treatment with or without FN for 72 h. Prior to fixation, cells were washed with PBS and then fixed 2.5% glutaraldehyde in 0.1 M PBS buffer. Following this, the cells were further fixed with 1% osmium tetroxide in 0.1 M cacodylate buffer. Specimens were stained with aqueous uranyl acetate and lead citrate and then observed using a Jeol-100 CX II (JEOL, Peabody, MA, USA) at 80 kV.

### 2.7. Clonogenic Assays

Cells were seeded in 12-well plates at a density of 400 cells per well for SKOv3-DIRAS3 and 100 cells per well for OVCAR8-DIRAS3 and cultured overnight at 37 °C. Then the cells were incubated with or without DOX to induce DIRAS3 expression for the first 3 days. The cells were then maintained with or without FN and the FAK inhibitor Defactinib for a total of 14 days. Crystal violet was used to stain the colonies. Plates were washed gently with water and air-dried. Colonies containing at least 50 cells were counted manually.

### 2.8. Sulforhodamine B Assay

Growth inhibition was evaluated by using the sulforhodamine B (SRB) assay. A total of 5 × 10^3^ cells were seeded in 96-well plates in triplicate and incubated overnight. Cells were then treated with 1 ug/mL of DOX to induce DIRAS3 expression, along with various doses of defactinib. After 5 days, cells were fixed with 30% TCA for 1 h at 4 °C. Plates were washed three times with water, air-dried, and stained for 30 min with SRB dye in 1% acetic acid. Plates were washed with acetic acid, followed by air-drying. SRB dye was solubilized with 100 μL of 10 mM Tris base, and absorbance was measured at 510 nm using a microplate reader.

### 2.9. Generation of Cell Lines for Monitoring Autophagy Flux

As previously described [24], the OVCAR8-DIRAS3 inducible cell line was transduced with a retrovirus carrying the pBabe-mCherry-eGFP-LC3 plasmid (C-G-LC3; Addgene 22,418; deposited by Jayanta Debnath). Following 18 h of transduction, the viral medium was replaced with complete-growth medium. Expression of eGFP and mCherry was confirmed after 48 h post-transduction. Cells were then incubated with selection media supplemented with 1 ug/mL puromycin and incubated for up to 72 h. Subsequently, cells were then sorted for mCherry and eGFP double-positive cells.

### 2.10. Statistical Analysis

Statistical analysis was performed using either Student’s *t*-test (two-sample, assuming unequal variances), one-way ANOVA, or two-way ANOVA. A *p*-value of less than 0.05 was considered statistically significant.

## 3. Results

### 3.1. Fibronectin (FN) Inhibits DIRAS3-Induced Autophagy

Our previous studies have demonstrated that matrix proteins, such as fibronectin, collagen, and poly-l-lysine, can partially rescue cells from DIRAS3-induced autophagic cell death in culture. To further investigate the effect of FN on DIRAS3-induced autophagy, we treated SKOv3-DIRAS3 and OVCAR8-DIRAS3 cells with FN. FN inhibited the induction of autophagy mediated by DIRAS3 in both cell lines, indicated by a decrease in the conversion of LC3I to LC3II (Figure 1A–D). Autophagy was also measured by fluorescence microscopy, where LC3B staining detected an increase in LC3 puncta after DOX-induced DIRAS3 re-expression in SKOv3-DIRAS3 ovarian cancer cells. Treatment with FN decreased LC3 puncta (Figure 1E), suggesting that FN inhibits DIRAS3-mediated autophagy. Furthermore, transmission electron microscopy detected classic autophagic vesicles with double membrane after 72 h treatment of DOX (to induce DIRAS3) in SKOv3-DIRAS3 and OVCAR8-DIRAS3 cells. Typical autophagosomes, characterized by multiple lamellae and digested material, were observed upon in the DIRAS3 re-expression, and this effect was reversed with FN treatment (Figure 1F,G). In a complementary approach autophagy flux was evaluated by using mCherry-LC3 fluorescence and p62 expression. mCherry-LC3 analysis showed that DIRAS3 induced autophagy flux as suggested by the increased in red fluorescence dots in cells treated with DOX. However, DIRAS3-induced autophagy flux was reduced after treatment with FN, suggested by increased in yellow dots, indicating a block in autophagic progression (Appendix A). Consistent with these findings, Western blot analysis of p62 demonstrated increased p62 levels following FN and DOX co-treatment (Appendix A). Together, these results indicate that fibronectin inhibits DIRAS3-induced autophagy.

### 3.2. FN Inhibits DIRAS3-Induced Autophagy by FAK Activation

To identify potential signaling pathways affected by FN and DIRAS3, we analyzed specimens with reverse-phase protein arrays (RPPAs). In SKOv3-DIRAS3 cells, the FAK/AKT pathway was significantly downregulated by DIRAS3 re-expression, with a decrease in pFAK (Tyr 397), pAKT (Ser 474) (Figure 2A,B). In addition to FAK and AKT, other growth and autophagy regulators were identified in the RPPA data. Our results show that DIRAS3, induced by DOX treatment, leads to an increase in Tuberin levels which correlates with a decrease in phospho-mTOR, indicating activation of autophagy, since mTOR is a well-known autophagy inhibitor. However, co-treatment with FN decreased the expression of Tuberin but did not restore mTOR phosphorylation, suggesting that DIRAS3 may suppress mTOR activity through both Tuberin-dependent and -independent mechanisms. This finding implies that the extracellular matrix can partially counteract DIRAS3 signaling yet is insufficient to fully reverse its effect on mTOR and autophagy activation. Given the important role of FAK in oncogenic signaling [25,26,27,28,29], we investigated the potential relationship between FAK and DIRAS3. To explore underlying mechanisms, we performed Western blot analysis to assess FAK and AKT phosphorylation. Treatment with FN increased phosphorylation of FAK and AKT in both SKOv3-DIRAS3 and OVCAR8-DIRAS3 ovarian cancer cell lines (Figure 2C,D).

### 3.3. DIRAS3 Induces Autophagy by Inhibiting the Integrin β1/FAK Signaling Pathway

We investigated DIRAS3’s inhibition of the FN receptor integrin β1. We treated cells with DOX and performed Western blot analysis. We found that DIRAS3 inhibited integrin β1 in a time-dependent manner in SKOv3-DIRAS3 and OVCAR8-DIRAS3 cells (Figure 3A,B).

### 3.4. FAK Inhibition Decreases Downstream Signaling and Enhances DIRAS3-Induced Autophagy

To determine whether decreasing FAK activity would increase autophagy, we used defactinib, selective FAK inhibitor. Defactinib selectively reduced the expression of pFAK (Tyr397) in SKOv3-DIRAS3 and OVCAR8-DIRAS3 cells (Figure 4A,B). Additionally, FAK inhibition restored the DIRAS3-induced conversion of LC3B-I to LC3B-II, which was suppressed by FN (Figure 4C,D) in both cell lines. 

### 3.5. Fibronectin Rescues DIRAS3-Induced Cell Death

To determine whether FN could reverse autophagic cell death induced by prolonged DIRAS3 expression, the effect of DIRAS3 expression and treatment with FN and defactinib was measured in clonogenic assays. SKOv3-DIRAS3- and OVCAR8-DIRAS3-inducible cells were treated with DOX to induce DIRAS3 expression in the presence and absence of FN. As shown in Figure 5A–D, treatment with FN significantly increased the number of colonies compared to the DIRAS3 re-expression group, indicating that FN can at least partially rescue DIRAS3-induced autophagic cell death. A combination of DIRAS3 expression and defactinib produced the greatest in inhibition, but this could be partially reversed with FN.

### 3.6. FAK Inhibition in Combination with Doxycycline Decreased Tumor Growth

We evaluated the effects of defactinib and DOX combination in vivo in human ovarian cancer xenograft models generated using SKOV3-DIRAS3 human ovarian cancer cells. As shown in Figure 6, mice responded well to both single-agent treatments and the combination. Tumor volume growth was significantly inhibited by the combination of DOX and defactinib treatment compared to control (**** *p* < 0.0001) or to defactinib (**** *p* < 0.0001).

## 4. Discussion

Autophagy is a cellular process that can be activated by different cellular stressors, such as nutrient deprivation, hypoxia, or chemotherapy. Interestingly, autophagy has a dual role as it can inhibit or sustain tumor growth depending on the cellular context [11,12]. In early stages, autophagy contributes to cellular homeostasis by degrading damaged organelles and proteins, reducing genomic instability and suppressing transformation. In contrast, in later stages of tumorigenesis, autophagy may act as a survival mechanism, allowing cancer cells to adapt under harsh conditions such as hypoxia and nutrient deprivation commonly found in the tumor microenvironment.

In a previous study, we documented the dual role of autophagy in the setting of DIRAS3-induced autophagy. Re-expression of DIRAS3 induced autophagic cell death in cultured ovarian cancer cells, consistent with autophagy tumor suppressor function. When, however, these cells are grown as xenografts, they remain dormant for several weeks following DIRAS3 induction, resuming growth promptly upon downregulation of DIRAS3, suggesting autophagy as a pro-survival mechanism. Treatment of dormant ovarian cancer cells with the autophagy inhibitor chloroquine prevented tumor outgrowth when DIRAS3 levels were reduced, suggesting that autophagy in dormant xenografts appears to be essential for cancer cell survival in a poorly vascularized and nutrient-poor microenvironment.

Elements of the extracellular membrane (ECM), such as FN, can rescue ovarian cancer cells from DIRAS3-induced autophagic death in vitro. Based on these observations, we explored the effect of FN on DIRAS3 expressing SKOv3 and OVCAR8 ovarian cancer cells.

Here, we demonstrated that re-expressing DIRAS3 induces autophagy in both SKOv3-DIRAS3 and OVCAR8-DIRAS3 ovarian cancer cells. Our findings indicate that FN inhibits DIRAS3-induced autophagy in ovarian cancer cells, as evidenced by reduced conversion of LC3I to LC3II on Western blots, decreased LC3 puncta observed by fluorescence staining, increased p62 expression, and impaired autophagic flux as demonstrated by mCherry-LC3 reporter analysis. To elucidate the mechanism by which FN decreased DIRAS3-induced autophagy, we utilized Reverse Phase Protein Array (RPPA). Among changes in signaling induced by DIRA3 expression, the FAK pathway was the most significantly affected, particularly involving phosphorylated FAK (p-FAK, Tyr 397) and AKT (p-AKT, Ser 474). FN decreased the inhibitory effect of DIRAS3 on both p-AKT and p-FAK.

Given the important role of FAK in oncogenic signaling, we further examined the potential relationship between FAK and DIRAS3 [29]. FAK can be activated by various growth factor receptors, G protein-coupled receptors, and integrins, triggering downstream signaling to a variety of target molecules involved in different cellular processes [29]. FAK is constitutively associated with the β1 subunits of integrin receptors. Binding of the ECM components to integrins leads to the activation of FAK. Our results demonstrate that the addition of fibronectin, a ligand for the αvβ1 integrin receptor, enhances FAK activity and diminishes DIRAS3-induced autophagy. These findings suggest that FAK is important in modulating DIRAS3-induced autophagy and that targeting FAK could be a promising strategy to enhance DIRA3-induced autophagic cell death in dormant cancer cells.

To evaluate the clinical relevance of FN1 and ITGB1 expression, survival analysis was conducted using data from the Human Protein Atlas (THPA) (Appendix A). Gene expression levels were stratified into high and low groups, and Kaplan–Meier survival curves were generated for overall survival (OS) or progression-free survival (PFS). High expression of FN1 was significantly associated with reduced survival (log-rank *p* = 0.004), while elevated ITGB1 expression also strongly correlated with worse prognosis (log-rank *p* = 0.00011). These results suggest that increased expression of fibronectin and integrin β1, key players in the DIRAS3-autophagy axis, may serve as adverse prognostic biomarkers in ovarian cancer.

Defactinib, a pharmacologic inhibitor of FAK, enhanced DIRAS3-induced autophagy in ovarian cancer cells cultured with fibronectin. Additionally, FN provided partial protection from DIRAS3-induced cell death in culture, judged by an increase in colony formation. Moreover, treatment with defactinib in the presence of FN significantly decreased the ability of cells to form colonies. Moreover, defactinib decreased tumor growth in the SKOV3-DIRAS3 xenograft model.

While our study provides valuable insight into the potential role of FN/FAK/AKT signaling in DIRAS3-mediated autophagy, this study has certain limitations. Our findings are based on two ovarian cancer cell lines, SKOv3 and OVCAR8, which cannot represent all the genotypic and diversity of all ovarian cancer subtypes. Therefore, future studies need to be performed using a broader range of ovarian cancer models, including patient samples, organoids, and genetically diverse in vivo models. Additionally, future in vivo studies will be designed to include the collection of tumor samples at different time points for the evaluation of autophagy markers. This data will provide a more comprehensive analysis of the regulation of DIRAS3-mediated autophagy and its regulation by the tumor microenvironment.

In conclusion, our findings reveal a complex interaction between DIRAS3, autophagy, and the tumor microenvironment, particularly the influence of fibronectin in modulating these processes. Our findings suggest that the extracellular matrix (ECM) plays an important role in coordinating the function of integrin β1, FAK, and AKT to regulate DIRAS3-mediated autophagy. This regulation is essential for the survival of the dormant ovarian cancer cells, which represent a challenge for effectively treating ovarian cancer. The persistence of these dormant, drug-resistant cells following primary treatment remains a major barrier to improving survival rates among women with ovarian cancer. Future studies will focus on further elucidating these interactions and explore potential therapeutic strategies to target and eliminate dormant ovarian cancer cells, ultimately improving patient outcomes.

## Figures and Tables

**Figure 1 cells-14-01250-f001:**
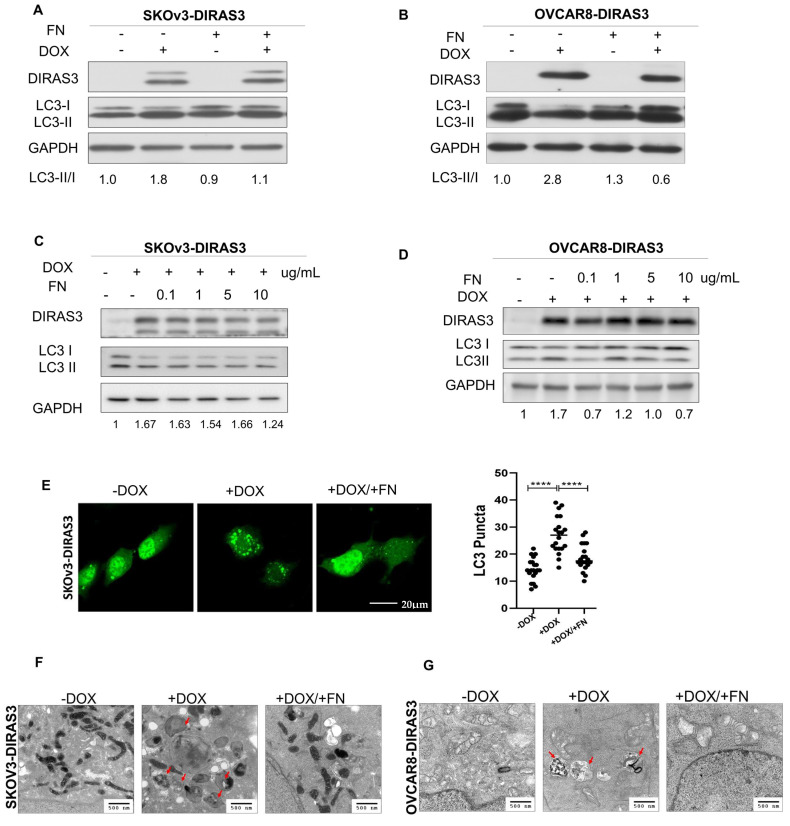
Fibronectin (FN) inhibited DIRAS3-induced autophagy. (**A**–**D**) Western blot analysis was performed to examine DIRAS3 and LC3 expression. SKOV3-DIRAS3 and OVCAR8-DIRAS3 cancer-cells were seeded at 1.5 × 10^5^ cells/well and treated with DOX (1 µg/mL) with or without fibronectin (FN) (0.1–10 ug/mL). Densitometry was performed with ImageJ. (**E**) Analysis of autophagy induction by expression of LC3 puncta. Cells were plated in chamber slides and treated with or without DOX (1 ug/mL) and with or without FN (10 ug/mL) for 48 h. LC3 puncta (green dots) were quantified to assess autophagy induction. **** *p* < 0.0001. (**F**,**G**) Transmission electron microscopy (TEM) analysis to measure autophagy. Cells were treated with DOX with or without FN for 48 h. Red arrows indicate typical autophagosomes with bilayer membrane.

**Figure 2 cells-14-01250-f002:**
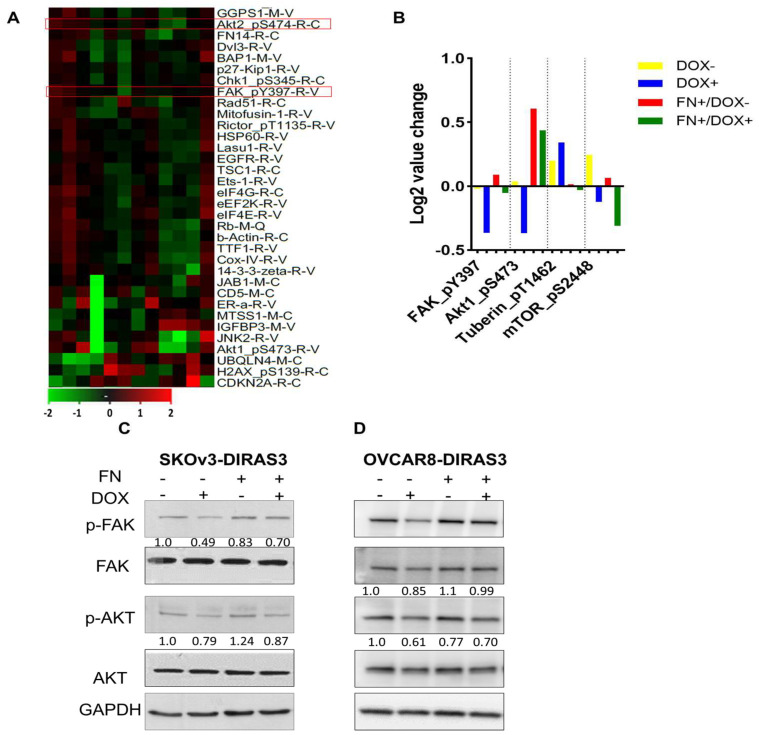
Fibronectin (FN) inhibited DIRAS3-induced autophagy by mediating the FAK/AKT signaling pathway. (**A**) Heatmap of reverse-phase protein array (RPPA) analysis. SKOv3-DIRAS3 cells treated with DOX (1 ug/mL) with or without FN (1 ug/mL) for 24 h. Red boxed indicate the proteins selected for further analysis. (**B**) Selected differentially expressed proteins between treatment groups. (**C**,**D**) Western blot analysis was performed to evaluate expression of the FAK/AKT signaling pathways after treatment with DOX +/− FN.

**Figure 3 cells-14-01250-f003:**
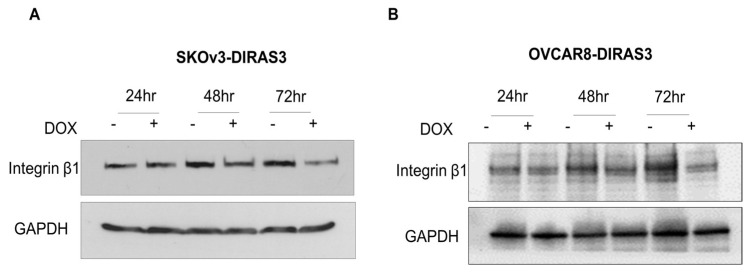
DIRAS3-induced autophagy by inhibiting integrin β1/FAK signal. (**A**,**B**) Western blot analysis after treatment with DOX during 24–72 h. Blots were probed to evaluate expression of β1. GAPHD was used as loading control.

**Figure 4 cells-14-01250-f004:**
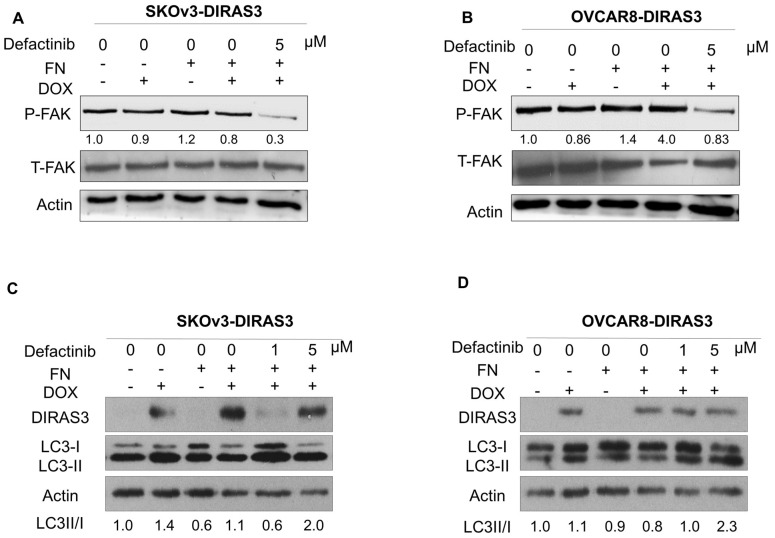
FAK inhibition decreases downstream pathway and increases DIRAS3-induced autophagy. (**A**,**B**). SKOv3-DIRAS3 and OVCAR8-DIRAS3 cells were treated with the FAK inhibitor defactinib (5 µM), with or without FN protein, and lysate was collected for analysis. Western blots were performed to evaluate the expression of phospho-FAK and total FAK. (**C**,**D**) SKOv3-DIRAS3 and OVCAR8-DIRAS3 cells were treated with the 1 or 5 µM of defactinib with or without FN. Western blot was performed to evaluate the autophagy marker LC3. Densitometry was calculated using Image J.

**Figure 5 cells-14-01250-f005:**
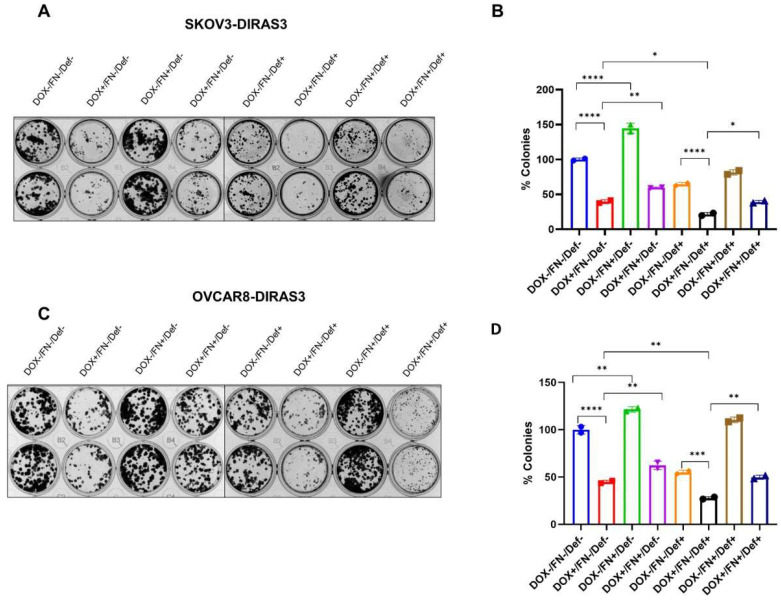
FN rescues ovarian cancer cells from DIRAS3-induced cell death. (**A**,**C**) Clonogenic assays and (**B**,**D**) quantification of colonies in SKOv3-DIRAS3 and OVCAR8-ARHI ovarian cancer cells that were treated with DOX, FN, and/or defactinib and allowed to grow for 14 days. Number of colonies were quantified manually and represented in the bar graph. The columns indicate the mean of colonies and error bars indicate SD. The statistical analysis was calculated using 1-way-ANOVA. * *p* < 0.05, ** *p* < 0.001, *** *p* < 0.001 and **** *p* < 0.0001.

**Figure 6 cells-14-01250-f006:**
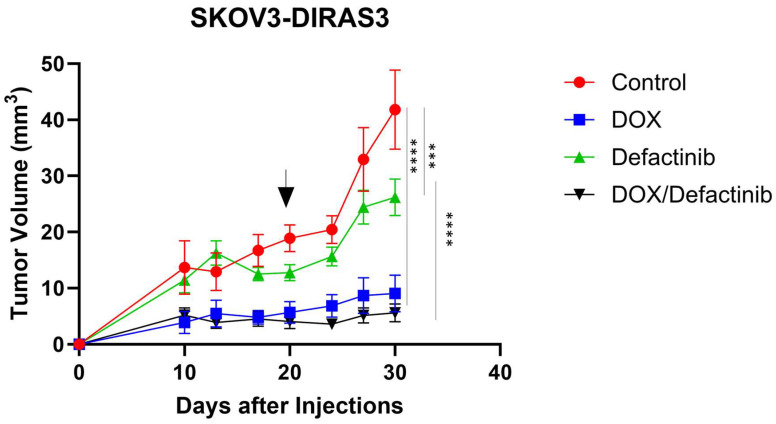
FN rescues DIRAS3-induced cell death in vivo. Tumor growth by volume of ovarian cancer xenografts in female BALB/c nu/nu after treatment with DOX and defactinib. Mice were injected subcutaneously with 5.5 × 10^6^ SKOv3-DIRAS3 cells. Tumor-bearing mice were randomized into 4 groups (n = 10). Mice were treated with defactinib (25 mg/kg orally twice daily) with or without the addition of doxycycline in their drinking water for the duration of the experiment. Graph was generated by GraphPad Prism 10.3.1. Two-way ANOVA and Tukey’s multiple comparison were used for tumor growth. Asterisks denotes significant difference (*** *p* < 0.001 and **** *p* < 0.0001). Arrow indicates the time when DOX was removed.

## Data Availability

The data generated in this study are available upon request from the corresponding author.

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
