# Peer review of "DIRAS3 Inhibits Ovarian Cancer Cell Growth by Blocking the Fibronectin-Mediated Integrin β1/FAK/AKT Signaling Pathway"

_cells, 2025, doi:10.3390/cells14161250_

Round 1

Reviewer 1 Report

Comments and Suggestions for Authors

This manuscript by Jing Guo investigates the role of fibronectin (FN) in counteracting DIRAS3-induced autophagic cell death, thereby promoting ovarian cancer cell survival. The topic is relevant and the experiments are generally well described. However, some issues should be addressed to improve the clarity and scientific rigor of the manuscript:

Figure 1A-D: The reported decrease in the conversion of LC3-I to LC3-II, used to support the conclusion that FN inhibits DIRAS3-mediated autophagy induction, is not clearly evident. To strengthen this point, the authors should include additional assays to quantify autophagic flux (e.g., p62 western blot; use of lysosomal inhibitors).

Figure 2B: This panel shows the expression of Tuberin and mTOR, yet these proteins are not mentioned or discussed in the main text. The authors should provide a description and interpretation of these results in the revised version of the manuscript to clarify their relevance.

In vivo experiments: The in vivo data are limited to tumor growth curves. To provide a more comprehensive characterization of the tumor response, I suggest including immunohistochemical analysis of tumor sections to assess FAK activation, autophagy markers (such as LC3 or p62), and proliferation markers (e.g., Ki-67). This would significantly enhance the impact and interpretability of the in vivo findings.

Author Response

Please see attachment with the point-by-point responses to the reviewer's comments

Reviewer 2 Report

Comments and Suggestions for Authors

The background is comprehensive but could benefit from a clearer linkage between DIRAS3, fibronectin, and clinical implications earlier in the text. Consider emphasizing the translational potential of targeting the FN/integrin β1/FAK/AKT axis in dormant ovarian cancer cells.

Figure 1: Include scale bars in images (E-G) for better interpretation.

Figure 5: Clarify statistical significance markers (e.g., asterisks) in the bar graphs for colony counts.

Elaborate on the dual role of autophagy in cancer (pro-survival vs. pro-death) in the discussion to contextualize why DIRAS3-induced autophagy shifts outcomes in vitro vs. in vivo.

Address potential limitations (e.g., generalizability beyond SKOv3/OVCAR8 models).

Page 2: “inhibits DIRAS3-induced autophagic death” → “inhibits DIRAS3-induced autophagic cell death” for consistency.

Author Response

Please see the attachment with the point-by-point response to the reviewer's comments

Round 2

Reviewer 1 Report

Comments and Suggestions for Authors

The authors have satisfactorily addressed my questions; therefore, in my opinion the manuscript is ready for publication